# Research on Laser-TIG Hybrid Welding of 6061-T6 Aluminum Alloys Joint and Post Heat Treatment

**Hongyang Wang \*** , **Xiaohong Liu and Liming Liu**

Key Laboratory of Liaoning Advanced Welding and Joining Technology & School of Materials Science and Engineering, Dalian University of Technology, Dalian 116024, China; Liuxh_72@163.com (X.L.); liulm@dlut.edu.cn (L.L.)
\* Correspondence: wang-hy@dlut.edu.cn

**Abstract:** The 6061-T6 aluminum (Al) alloys was joined by the laser induced tungsten inert gas (TIG) hybrid welding technique. It mainly studied the influences of welding parameters, solution, and aging (STA) treatment on the microstructure and tensile properties of Al alloy hybrid welding joints. Microstructures of the joints were also analyzed by optical microscopy and transmission electron microscopy. Results showed that the laser induced arc hybrid welding source changed the microstructure of the fusion zone and heat effect zone. STA treatment effectively improved the mechanical properties of the softening area in the hybrid welding joint, whose values of the tensile strength and elongation were on average 286 MPa and 20.5%. The distribution of the reinforcement phases and dislocations distributed were more uniform, which improved the property of STA treated joint.

**Keywords:** Al alloy; laser induced arc hybrid welding; heat treatment; microstructure

## 1. Introduction

Aluminum alloys are used as structural materials, due to the high strength, excellent corrosion resistance, and low density, which makes them promising alternatives to steels for lightweight structures such as vehicles and airplanes [1–7]. Aluminum alloys are also extensively used in aviation, vessel, and electronics industries due to their light weight, high specific strength, good corrosion resistance, and excellent mechanical properties [8–16]. Aluminum profile heat treatment is one of the most important processes in the production of materials. Compared with other processes, heat treatment generally does not change the shape of the specimen and the overall chemical composition, but rather provides or improves the performance of the specimen by changing the internal microstructure of the specimen, or the chemical composition of the surface of specimen. As a heat treatable alloy, 6061-T6 had found a wide application in the fabrication of lightweight structures. It was excellent weldability over other high strength aluminum alloys [17].

Lots of welding methods were used in the joining of aluminum alloys such as tungsten inert gas (TIG), metal inert gas (MIG), laser beam welding (LBM), friction stir welding (FSW), hybrid welding process, etc. Ahmad R et al. [18] investigated the effects of three filler wires, namely, ER5356, ER4043, ER4047, on the microstructures and properties of a butt welded joint of AA6061 aluminum alloy by TIG welding. Kulekci et al. [19] compared the mechanical properties of welded joints of AW-6061-T6 aluminum alloy gained by FSW and MIG, respectively. They pointed out that the joint obtained with the FSW welding process had better mechanical properties and a narrower heat affected zone than those obtained by the MIG process. Narsimgachary et al. [20] welded a 6061-T6 aluminum alloy by the laser welding process and studied the temperature profile on the microstructure and mechanical properties of the joint. Chen Zhang et al. [21] used 5–6 kW laser-arc hybrid welding source to weld an

AA6082 aluminum alloy, and pointed out that the hybrid welding had some advantages, compared with the joints welded by simple laser or pure arc, such as the higher fatigue limit, lower percent porosity, grains refinement, etc.

Since the 6061-T6 aluminum alloy can be strengthened by the heat treatment process, several researchers applied post-weld heat treatment processes to improve the properties of the welded joints. Mohammad et al. [22] studied the influence of solution treatment and subsequent artificial aging treatment on the joint of AA6061-T651 aluminum alloy by the TIG welding process. The results showed that post-weld heat treatment significantly improved the mechanical properties of the joint, due to grain refinement and precipitation hardening. Dong peng et al. [23] analyzed the effects of aging treatment and heat treatment on the microstructure and mechanical properties of TIG-welded 6061-T6 aluminum alloy joints, and they pointed out that the mechanical properties of the joints were improved significantly because of distributing of few precipitates in the welded seam. Elangovan and Balasubramanian [24] investigated the effect of various post-heat treatments, namely, solution treatment, aging treatment, and the combination of them on tensile properties of FSW-ed AA6061 aluminum alloy.

With the development of the Al alloy manufacture, the requirement of the Al welding joint was not only the tensile strength, but also the plastic deformation capacity. It was very important for the multilink manufacture process, when the welding process was in the midst link but not the final process. The tensile strength of the above welding joints was high enough, but the ductility of the joints still needs to be improved. The ductility of the joints was mainly influenced by the microstructure, precipitate, and the dislocation of the joint. The low power laser induced arc hybrid welding source changed the heat distribution of the fusion zone, which made an obvious effect on the microstructure of the joint. Therefore, the ductility of the Al joints could be improved in laser induced arc hybrid welding process. Therefore, the low power pulsed laser induced TIG hybrid welding source was used to weld a 6061-T6 aluminum alloy in this essay. It was made to explore the influence of hybrid welding parameters and PWHT treatment on the tensile strength and ductility of the hybrid welding Al alloy joint.

## 2. Experiments

Commercial rolled 3 mm thick 6061-T6 aluminum alloy plates were used as base metal (BM), and machined into the size of $100 \times 50$ mm. The ER5365 filler wire with diameter of 1.2 mm was used in the experiments. Their chemical compositions are shown in Table 1. Figure 1 is the optical microstructure of BM. It was found that the grains in BM were elongated along the rolling direction, the grey phase was the matrix of $\alpha$ (Al), and black and white precipitate phase were $\beta$-$Mg_2Si$ and Al-Fe-Si, respectively. Table 2 presents the main mechanical properties of BM [2].

**Table 1.** Chemical compositions (wt%) of base metal and filler wire.

| Elements | Mg | Si | Fe | Zn | Cu | Al |
|---|---|---|---|---|---|---|
| 6061-T6 (BM) | 1.00 | 0.55 | 0.36 | 0.01 | 0.26 | Bal |
| ER5356 (filler wire) | 5.10 | 0.20 | 0.20 | 0.10 | 0.01 | Bal |

**Table 2.** Mechanical properties of BM.

| Material | Ultimate Tensile Strength (MPa) | Elongation (%) | Vickers Hardness (Hv) |
|---|---|---|---|
| 6061-T6 (BM) | 340 | 19 | 115 |

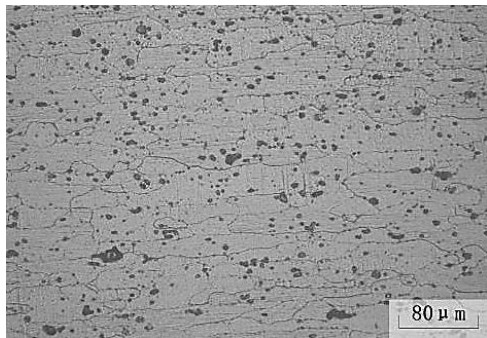

**Figure 1.** Microstructure of aluminum (Al) alloy BM.

The Al alloy butt joints are welded by a laser induced arc hybrid welding source and the schematic diagram of the welding process is shown in Figure 2. In the experiment, $\alpha$ is the angle between TIG touch and horizontal, $\beta$ is the angle between filler wire and horizontal, d is the distance between the laser incident point and filler wire on the plate, and $D_{la}$ is the distance between the laser incident point and tungsten electrode on the plate. The main welding parameters are shown in the Table 3. The pulse laser is used in the experiments. The pulse duration is 2 ms.

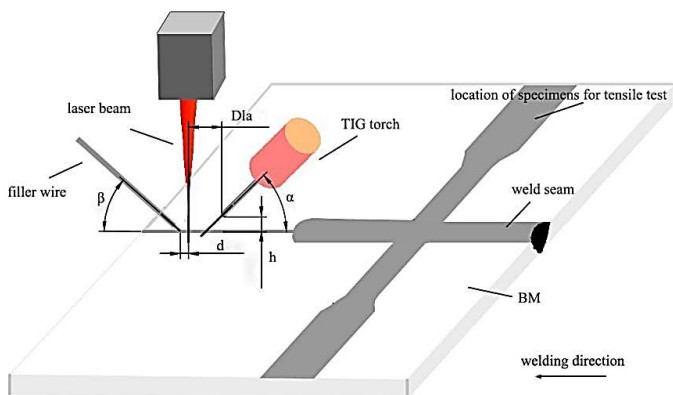

**Figure 2.** Laser induced arc hybrid welding process and the schematic diagram.

**Table 3.** Main parameters of laser induced arc welding Al alloy processes.

| Arc Current (A) | Average Laser Power (W) | Welding Speed (mm·min$^{-1}$) | Pulse Repetition Rate (Hz) | Peak Power (W) |
|---|---|---|---|---|
| 100–150 | 800–1000 | 500–700 | 40 | 6000~8500 |
| β (°) | h (mm) | d (mm) | α (°) | $D_{la}$ (mm) |
| 50 | 1.5 | 1.5 | 45 | 2 |

To understand the influence of post-weld heat treatment on the mechanical properties of the joints, the specimens were divided into two categories, namely, hybrid welded (HW) joints and solution treated and aged (STA) joints. In the process of STA treatment, the joints were treated in the furnace at 475, 520, and 565 °C for 1 h, quenched in a cold water bath and then were kept in the induction furnace at 155 °C for 8 h. The property of the STA joint with 520 °C treatment showed the best results, thus only the STA joint with 520 °C treatment was discussed in the article.

To evaluate the mechanical properties of specimens, BM, HW, AG, and STA joints were machined into the required dimensions according to the ASTM E8 E8M-11 standard. The tensile test was carried out at room temperature and a speed of 2 mm/min by using an electro-mechanical controlled universal testing machine. Three samples were tested for each treated joints and BM, and the average values were calculated to evaluate the tensile properties of the specimens.

The joints were sliced into a small suitable size, grinded, and polished according to the metallographic specimen preparation standard, etched by Keller reagent (1.5 vol% HF + 1.5 vol% HCl + 2.5 vol% $HNO_3$ + 95 vol% $H_2O$) for 1 min, and then observed by an optical microscope (OM). Hardness test was performed on the cross section of the joint. The distance between each two test points is 0.25 mm. Scanning electron microscopy (SEM) was used to character the fracture surfaces of the tensile specimens and understand the failure patterns of the specimens. The phase compositions of welding joints fractures were identified by a PANayltical (Almelo, Netherlands) Empyrean powder X-ray diffractometer. The strengthening phases and dislocations were observed by transmission electron microscopy (TEM), and the TEM samples were prepared by double-jet electrolytic polishing in an electrolyte (30 vol% nitric acid and 70 vol% methanol) at a temperature of −35 °C with a voltage of 20 V.

## 3. Results

### 3.1. Welding Process

In order to investigate the influence of the main welding process parameters on the geometrical dimensions of the weld seam, the influence of the laser beam power, arc current, and welding speed on the welding joint morphology are shown in Figure 3.

| Number | P I V (W) (A) (mm·min⁻¹) | Welding Joint Morphology | Cross Section of Joints |
|---|---|---|---|
| 1 | 750 130 500 | | |
| 2 | 750 150 550 | | |
| 3 | 750 160 600 | | |
| 4 | 800 130 550 | | |
| 5 | 800 150 600 | | |
| 6 | 800 160 500 | | |
| 7 | 900 130 600 | | |
| 8 | 900 150 550 | | |
| 9 | 900 160 500 | | |

**Figure 3.** Morphology and cross section of welding joints in different parameters.

It could be found that the arc current has the most significant effect on the weld geometry, compared with the laser beam power and welding speed. Except the arc current, the welding speed was another factor that influenced the reduction of the geometry of the weld seam, which had a significant effect on the front height and back height. Compared with the other two parameters, the laser beam power had a smaller effect on the weld geometry, but it had a greater effect on the weld width, which was higher than the other two factors.

The best parameters of the laser induced arc welding 3 mm thick Al alloys are shown in Table 4. The welding appearance and macrostructure of the joint are shown in Figure 4. The welded joint was obtained in the free formation state. It could be found that the welding appearance was uniform and no hump defects. There was no welding porosity and the cracks in the laser induced arc hybrid welding Al alloys joint. The width of welding beam was about 5 mm on top of the joint and 3 mm at the bottom of the joint. The heat effect zone of the joint was less than 2 mm.

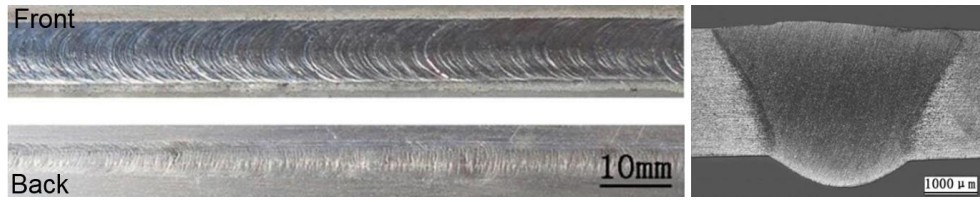

**Figure 4.** Appearance and the macrostructure of hybrid welding Al alloys joint.

**Table 4.** Best parameters for 3 mm thick Al alloys hybrid welding process.

| Arc Current (A) | Average Laser Power (W) | Welding Speed (mm·Min⁻¹) | Pulse Repetition Rate (Hz) | Pulse Peak Power (W) | Filler Speed m/Min |
|---|---|---|---|---|---|
| 150 | 900 | 600 | 40 | 8000 | 2 |

*3.2. Tensile Properties*

In order to understand the property of the hybrid welding joint, the tensile tests of the joints were put forward in different conditions. Table 5 shows the results of the transverse tensile test. The base metal (BM) showed a tensile strength and elongation of 340 MPa and 19%, respectively. However, the tensile strength and elongation of the hybrid welding (HW) joint were 234 MPa and 15.2%, respectively.

**Table 5.** Transverse tensile properties of laser–arc hybrid welded 6061-T6 joints.

| Joints | Ultimate Tensile Strength (MPa) | Elongation (%) | Joint Efficiency (%) | Fracture Positions of Tensile Specimens |
|---|---|---|---|---|
| BM | 330~340 | 18.7~19.4 | 100 | BM |
| HW | 230~240 | 14.30~15.71 | 66.8~70.8 | HAZ |
| AW (155 °C, 8 h) | 235~245 | 13.8~16.1 | 71.2~74.2 | HAZ |
| STA (520 °C) | 283~288 | 19.01~22.02 | 83.2~87 | FZ (Fusion zone) |

Firstly, the HW joints were treated with only artificial aging (155 °C, 4/8/12 h), but the only ageing treatment had no obvious improvement in the static tensile properties of the joints, thus it was not discussed.

As it was fractured on the HAZ of the HW joint, the solution treatment and ageing (STA) was done in different conditions. The STA treatment offered the obvious improvement on the mechanical properties of the joints. The values of the tensile strength and elongation of the STA joints were on average 286 MPa and 20.5%, which were 22% and 34% greater than those of HW joints, respectively. It should be pointed out that the fracture locations of HW joints were all in the HAZ, however the solution treated and aged HW joints failed in the fusion zone (FZ). The ratio of the tensile strength of the joints to the tensile strength of base metal was known as the joint efficiency [25–29]. The single

HW joints exhibited joint efficiencies of 68.82%, respectively, while the STA joints showed a high joint efficiency of 83.2%~87%.

### 3.3. Hardness

The Vickers hardness of the 6061-T6 aluminum alloy was about 110–120 Hv. Due to the heat resource acting on the center of the weld, the heating cycles for both sides of the weld suffered were the same during the welding process, thus the hardness distributions of the two sides should be similar. Figure 5 shows the hardness distribution of the hybrid welded joints under different treatments. The average hardness of the fusion zone was about 60–70 Hv. The lowest the hardness 53 Hv was observed in HAZ. After aging treatment, the hardness of each zone increased a little. However, the location of the lowest hardness was also in the HAZ.

As for the STA treatment joints, the change of the hardness of the fusion zone was similar to that of aged joints and the average hardness was 70 Hv. It was also noteworthy that the hardness of the STA joint had been greatly improved in the heat effect zone. Due to the great change of microstructure, the hardness of the STA treatment BM was lower than that of the original BM.

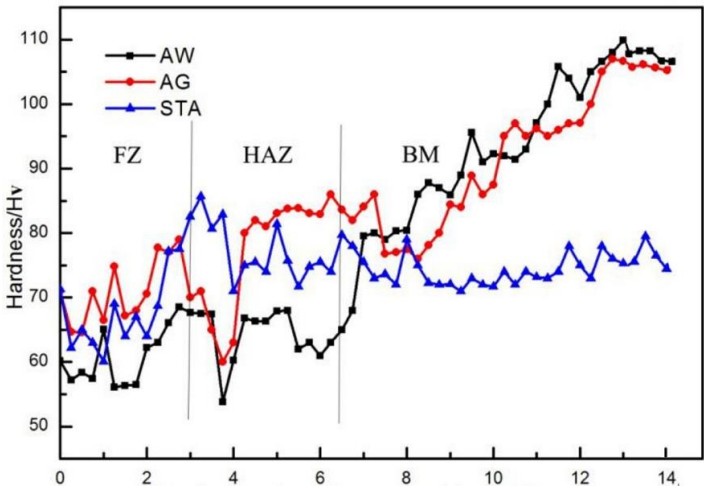

**Figure 5.** Hardness distributions of different treated joints.

### 3.4. Microstructure

The hybrid welded joints were initially observed by an optical microscopy and the microstructure of different zones of the HW joint are shown in Figure 6. The joint consists of three zones, namely, a fusion zone (FZ) (Figure 6a), a partial fusion zone (PFZ) (Figure 6b), and a heat effect zone (HAZ) (Figure 6c). The FZ was composed of a large number of equiaxed crystals and a small amount of columnar crystals. In the PFZ, there were plenty of columnar crystals along the direction that was perpendicular to the fusion line. Aluminum alloys have a high thermal conductivity, therefore the welding heat can quickly spread to HAZ, resulting in the microstructure of this zone (Figure 6c) being completely different from that of base metal. The soften area of the joint was formed because of the grain coarsening and the reversion of strengthening $\beta''$ (needle-like $Mg_2Si$) precipitates, thus the tensile tested specimens failed in this region. Plenty of nonhardening $\beta$ phases (flaky $Mg_2Si$, black precipitates in Figure 6c) can be found in HAZ, which are formed with the effect of welding thermal cycle.

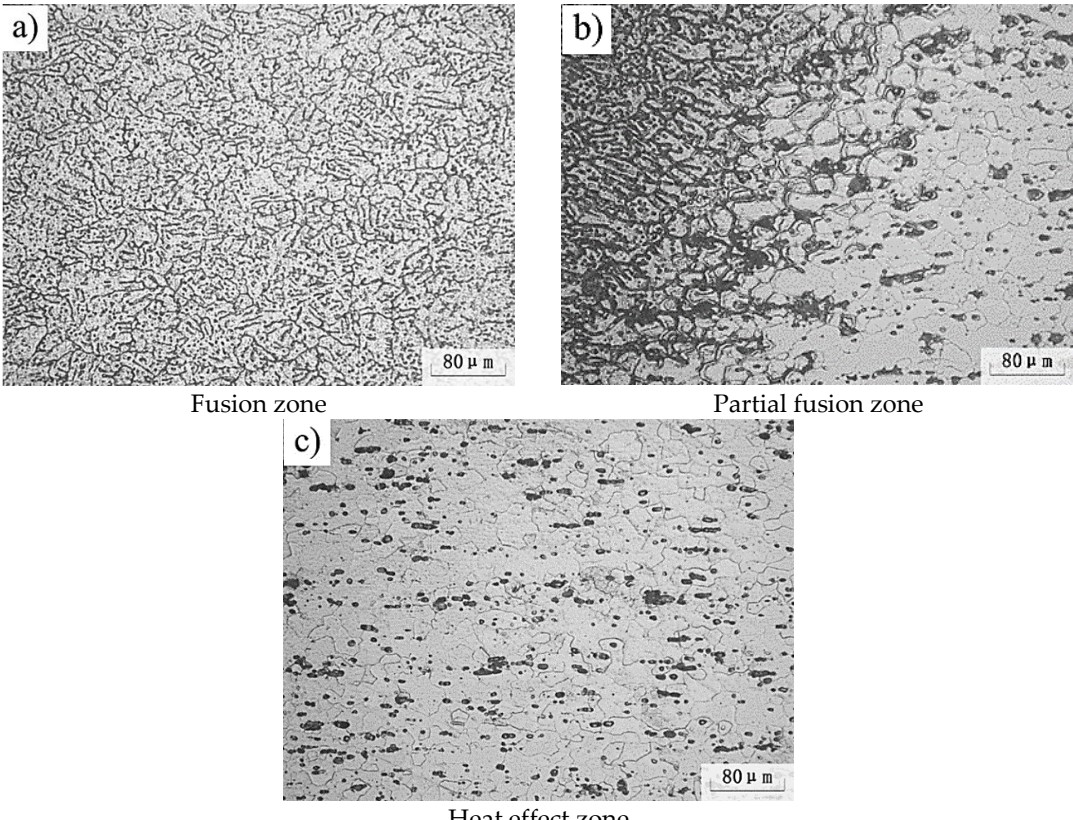

Figure 6. Optical microscopy of hybrid welding 6061-T6 joints.

The different regions of HW joint under STA treatment are presented in Figure 7. Compared with the corresponding area of the as-welded and artificial aged joints, the grain size and morphology had no obvious change, and the number of large size black precipitates was reducing and distributed more uniformly in the $\alpha$ (Al) matrix. However, it did not mean that the microstructure of the HAZ was not changed, as it was not observed simply by the OM test.

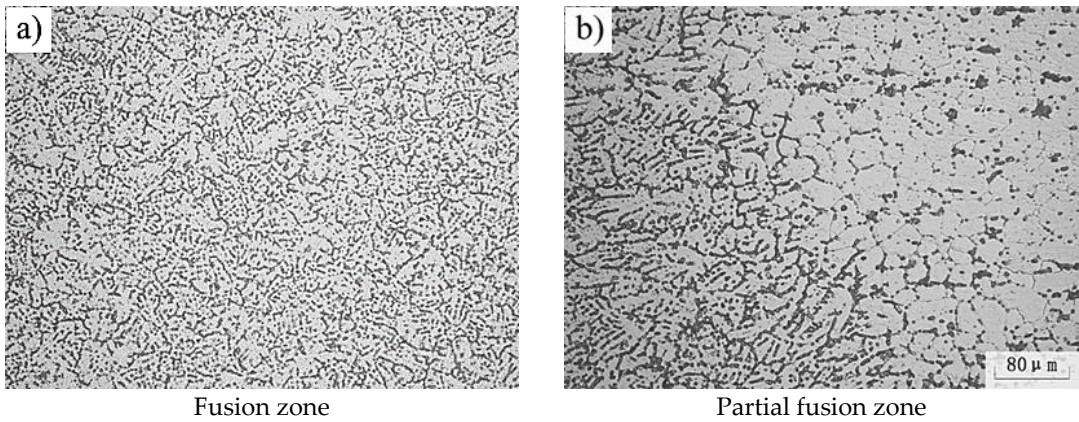

**Figure 7.** *Cont.*

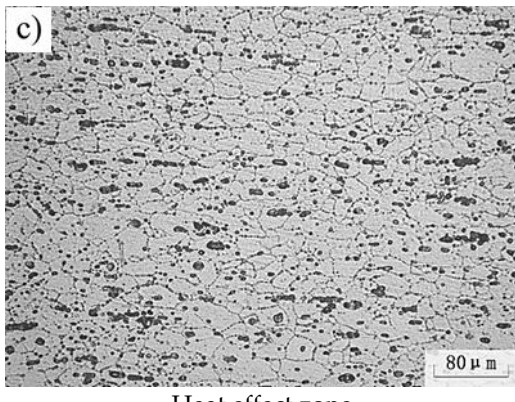
Heat effect zone

**Figure 7.** Optical microscopy of solution treated and aged 6061-T6 joints.

### *3.5. Fracture*

The failure patterns of the tensile tested specimens are characterized by SEM, as shown in Figure 8. A large number of dimples were found in all the fracture surfaces, which indicated that most of the failure was the result of ductile fracture. It can be observed that the dimples on the fracture surface of the STA joints (Figure 8b) are smaller than those of HW joint (Figure 8a). The experimental results were consistent with the conclusions offered by Hu and Richardson [30], which reported that there were a large number of fine dimples on the surface of the solution heat-treated samples. In the tensile process of ductile material, voids often formed prior to necking. If the necking was generated earlier, the formation of voids would become more dominant. The coarse and elongated dimples could be seen in the welded joints (Figure 8a) [28]. Figure 8c shows the XRD analysis results of the STA joint fracture, and $Mg_2Si$ is found on the fracture.

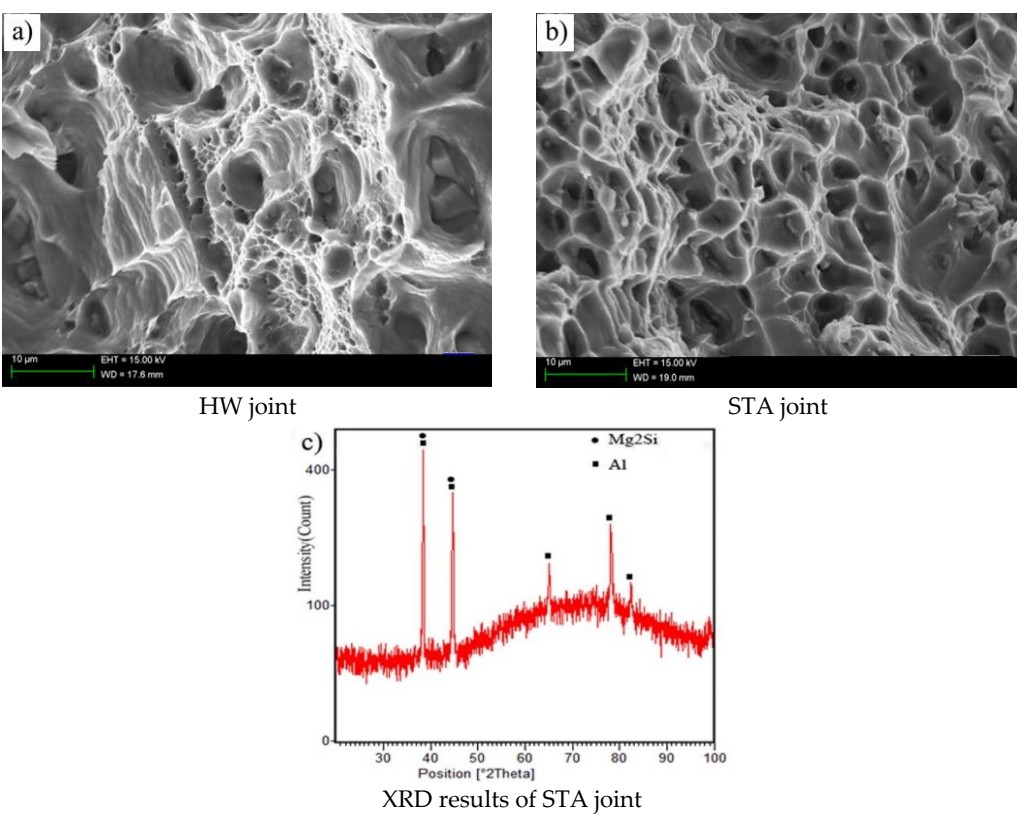
HW joint                STA joint
XRD results of STA joint

**Figure 8.** SEM and XRD of tensile tested specimen fractures.

### 3.6. Discussion

Figure 9 shows the distribution and shape of the strengthening precipitates in the FZ and HAZ under different heat treatment conditions. Mg$_2$Si is the major precipitate in 6000 series aluminum alloys. In the solution and aging (STA) joint, according to the different conditions (temperature), precipitation phases with different morphologies and strengthening effects would be sequentially precipitated. The precipitation process is generally: Supersaturated $\alpha$ solid solution → G·PI region (spherical objects without independent lattice structure) → G·PII region (needle-like ordered structure β″) → β′ (rod-like semicoherent structure) → β (flaky equilibrium phase Mg$_2$Si).

The coherent precipitates known as β″ (needle-like Mg$_2$Si) made the most strength effect on the mechanical property of the BM. A lot of tiny needle-shaped β″ precipitates uniformly distributed through the base metal (Figure 9a,b). In the low-magnification overview, only a small amount of nanometer precipitates could be seen, which illustrated that most of Mg$_2$Si in the BM exists in the form of β″.

Due to the direct effect of the hybrid welding source and the feeding of ER 5356 filler wire, the component and precipitates of the joints had been modified significantly in the FZ, and the precipitates state had been also changed in the welding process in HAZ. At low magnification, no β″ is found in the FZ of HW joint (Figure 9c). Compared with the base metal, the majority of β″ precipitates are transformed into a weak hardening rod—shaped β′ precipitates or the equilibrium flake β phases in the HAZ (Figure 9d), therefore some large particle black β phases can be observed in Figure 6c. As for simple artificial aged (AG) joints, a small quantity of precipitates agglomerated in the FZ (Figure 9e), and the density of the precipitates was higher than that of HW joint HAZ (Figure 9f). In the FZ and HAZ, most of the alloying elements were existing in the equilibrium strengthening phase. Therefore, it was difficult for the precipitation of strengthening β″ phase.

After the HW joint was treated by STA, a lot of nano-sized precipitates can be found in Figure 9e and f. Plenty of β″ were uniformly distributed in the FZ and their average width and length were about 9 and 190 nm, respectively. A great number of precipitates were uniformly dispersed in $\alpha$ (Al) matrix in the HAZ, whose size was varied between 25 and 130 nm. In the hybrid welding process, the Al alloy was fully melted, and the precipitate size was mainly decided by the solidification rate. However, in different parts of the laser induced arc hybrid welding fusion zone, the solidification rate was obviously different, thus the size of precipitate was changed in extent.

In the process of solid solution treatment, the alloy elements were dissolved into $\alpha$ solid solution as much as possible. Therefore, in the subsequent aging process, the precipitates were much more uniformly distributed than that under HW joint or aged state. At the same time, the content of Mg element in feeding of ER5356 filler wire was higher than that of BM, and it would provide a favorable condition for the formation of β″ precipitate, thus the content of precipitates in the FZ was more than that in HAZ.

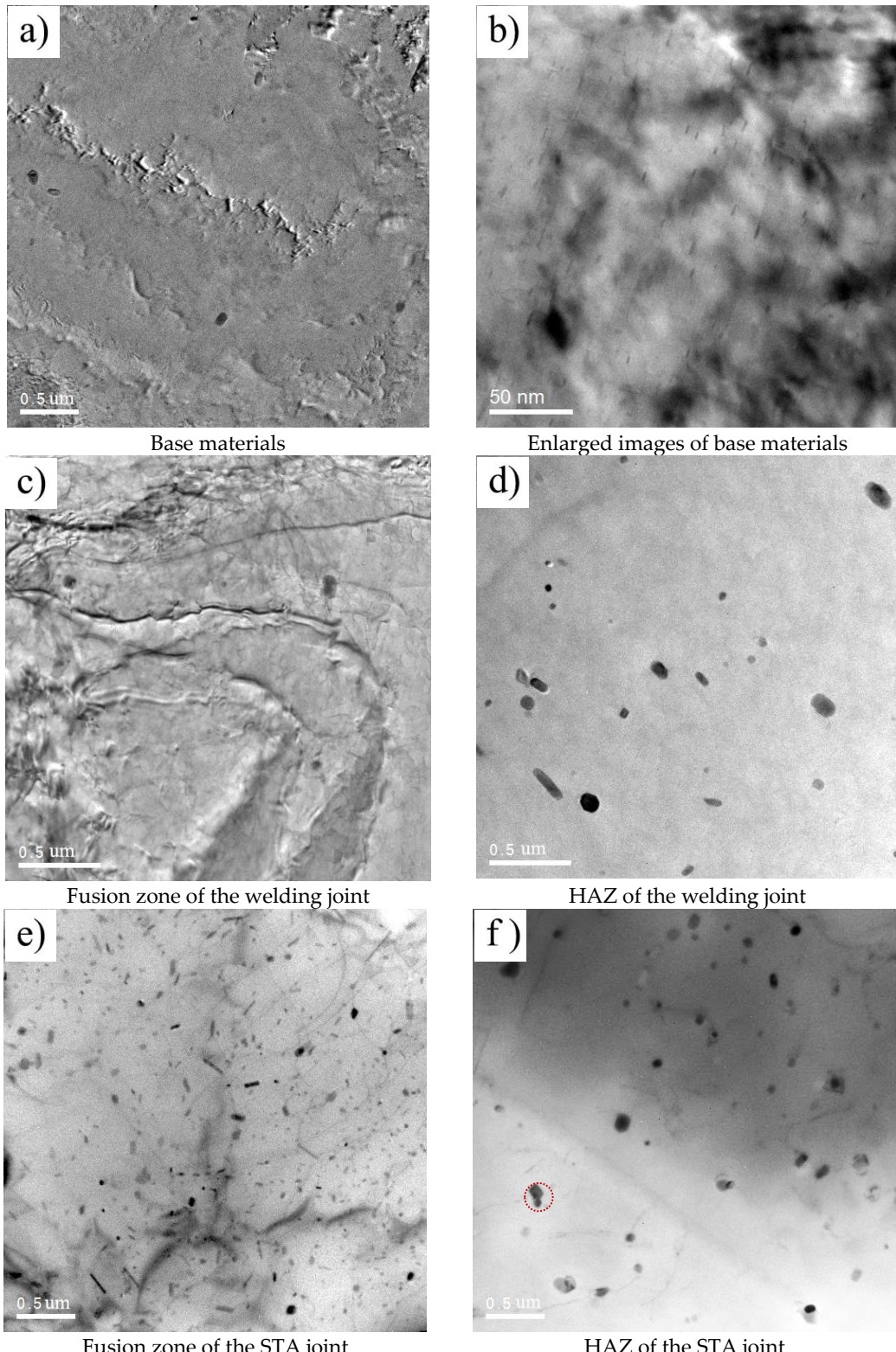

**Figure 9.** Distribution of β″ and in β in 6061-T6 BM and hybrid welded joints.

Figure 10 shows the spectral analysis results of precipitated phases in Figure 9d. Through the analysis of the organization of different regions, it was found that the elemental composition of this region was mainly composed of element Mg and Si. The element Cu was found, because of the influence

of copper sample stage. Therefore, it could be deduced that the precipitated phases in this area was mainly the Mg$_2$Si phase [24].

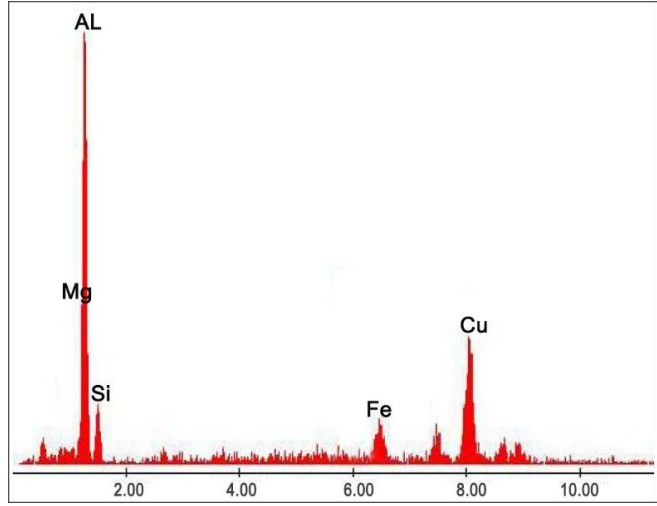

**Figure 10.** Spectral analysis results of precipitated phases in Figure 9d.

Figure 11 shows the dislocations structures of the BM and HW joints treated by different processes. Since the BM is in the rolling state, the dislocation density inside the BM grains was very high. The density of the dislocation was decrease in the hybrid welding joints (Figure 11a,b), especially in the HAZ of hybrid welding joint, where it did not find the dislocation tangling cells, but instead the dislocation stripes. In STA joints, the dislocation fringes were transformed into a single dislocation fringe with a more uniform distribution (Figure 11d,e). Therefore, the dislocation in STA joint had been rearranged in some extent.

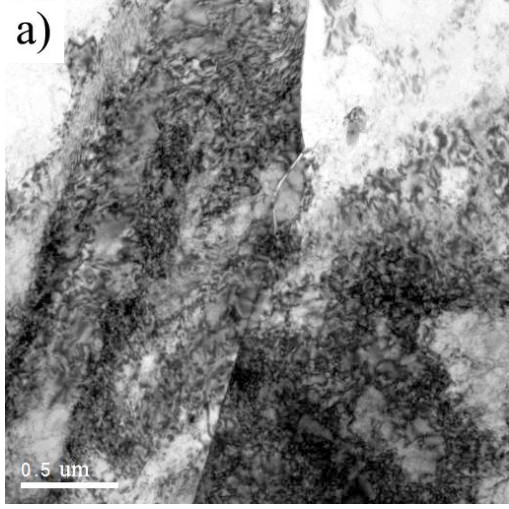

Base metal

**Figure 11.** *Cont.*

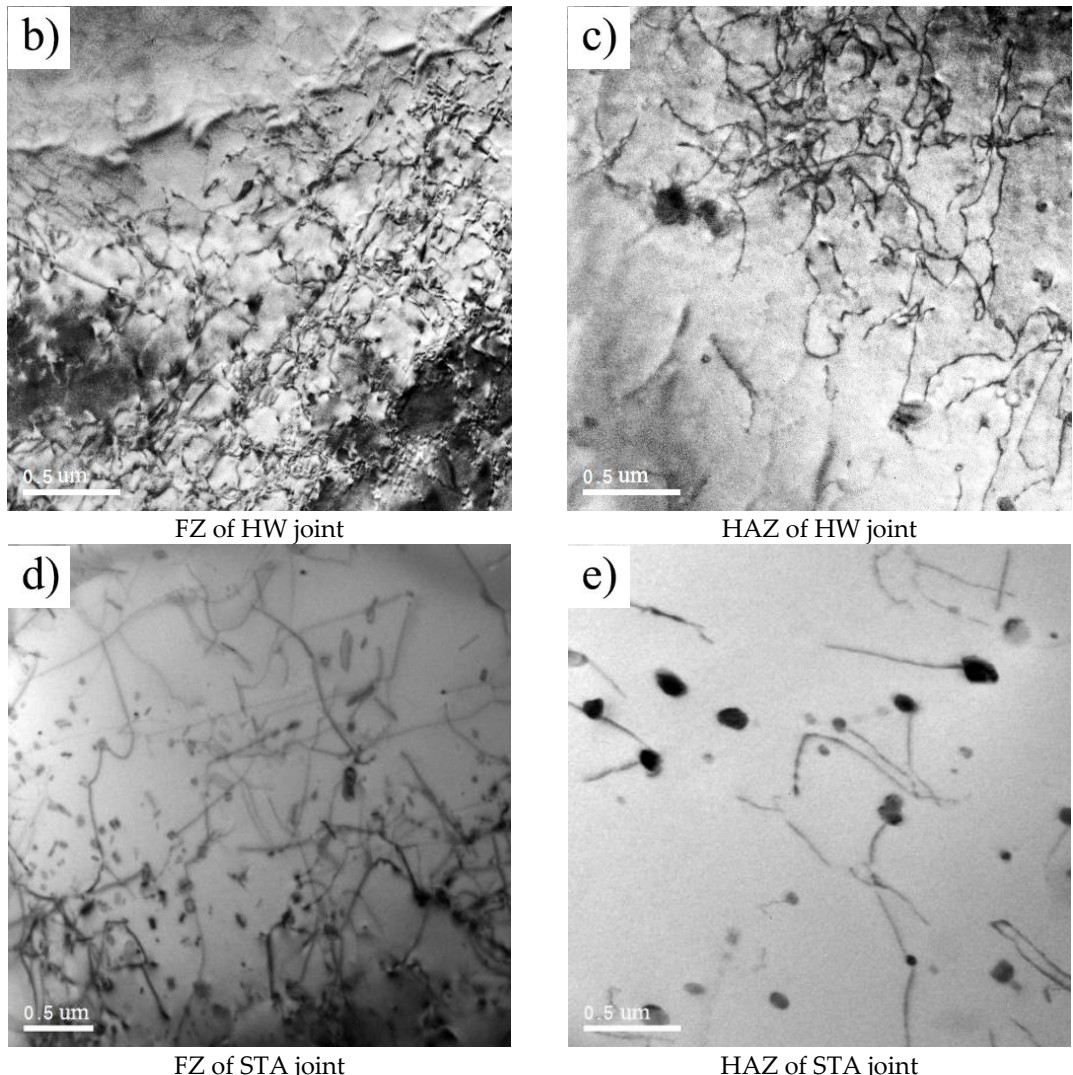

**Figure 11.** Dislocation structures in 6061-T6 hybrid welded and solution and aging (STA) joints.

From the experimental results, it was found that mechanical properties of HW joints were lower than those of the base metal. The failure location was about 4 mm from the center of the weld, where it corresponded to the position of the lowest hardness. With the effect of welding thermal cycles, the grains were coarse in the HAZ of HW joint (as shown in Figure 6c), and the β″ precipitates transformed into no or weak hardening β′ or β phases (Figure 9d). It became the crack sources in the process of tensile tests, therefore, a weak area was formed at the HAZ of HW joint, causing the specimen to failure.

In laser induced TIG hybrid welding Al alloy joint, the welding heat input was obviously lower than that in the TIG welding process. The microstructure in the fusion zone of HW joint was the fine equiaxed grain, whose size was evidently smaller than that in the BM. In the STA process, the grain size of the fusion zone was not changed, which exerted a beneficial effect on the property of the joint. Therefore, it could be deduced that: (1) After STA treatment for HW joint, there is no significant change in grain size in the FZ and it is helpful for the property of the joint; (2) the dislocation configurations in the corresponding regions of the HW joints are different, and the dislocations are changed after STA treatment, which is re-uniformly arranged due to recovery softening; (3) after STA treatment, nano-level strengthening phase β″ with uniform distribution appears in the joint, which plays a dominant role for the property improvement of the STA joint.

In the heat treatment strengthening aluminum alloys, the properties of the joints were determined by the existing forms, quantity, and distribution of the strengthening phases and also related to the density and distribution of the dislocations [31]. The diffuse precipitation precipitates strengthen and the density of dislocations reduction made an obvious influence on the comprehensive properties of materials. In addition, the uneven distribution of dispersed precipitates and dislocations in the HW joint would lead to stress concentration on HAZ in the tensile test process, resulting in uneven deformation of the specimens in each part. Therefore, the properties of the HW joint would be reduced because of the strengthen phase precipitation and stress concentration.

For the STA joints, the dispersion strengthening of $\beta''$ phase played a dominant role, thus the tensile strength and ductility of the STA joints were improved. In addition, the dislocations density in the FZ and HAZ of STA joints were changed to a similar level. Still the grain size of the FZ was slightly smaller than that of the base metal, and the grain size of the HAZ was nearly same with the BM. Therefore, the degree of deformation during the tensile test process gradually become uniform, and the elongation of the STA joint has been improved obviously. Due to lower tensile properties of the filler wire, the STA joint all failed approaching the fusion zone.

## 4. Conclusions

In this research, the influences of welding parameters and STA treatment on the microstructures and mechanical properties of the laser induced TIG hybrid welded 6061-T6 joints have been investigated and the main conclusions can be derived as follows:

The arc current made an obvious effect on the welding joint morphology. The microstructure of the laser induced arc hybrid welding Al alloy joint was mainly composed of the fine equiaxed grain and a small amount of columnar crystals, which made a beneficial effect on the STA treatment process. The aging treatment and solution and aging treatment made little effect on the grain size of the hybrid welded joint.

The tensile property of laser induced arc hybrid welded joint was 69% of the base metal, which was mainly influenced by the reduction of $\beta''$ phase. After the solution and aging (STA) treatment, the tensile property of the STA treated joint was 84% of the base metal, and the elongation of the joint was same as the base metal. The tensile strength and the ductility of the laser–arc hybrid welded Al alloy joints were improved by the hybrid effect of grain size refinement in the hybrid welding fusion zone and the STA treatment. After STA treatment, the nano-level strengthening phase $\beta''$ with uniform distribution appeared afresh in the joint, which played a dominant role for the property improvement of the joint.

**Author Contributions:** H.W. Conceptualization, Methodology, Supervision, X.L. Investigation, and L.L. Formal Analysis. All authors contributed to the writing and editing of the manuscript. All authors have read and agreed to the published version of the manuscript.

**Funding:** The authors gratefully acknowledge the support of the National Natural Science Foundation of China (U1764251 and 51975090).

**Conflicts of Interest:** The authors declare no conflict of interest.

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
