# Peer review of "Research on Laser-TIG Hybrid Welding of 6061-T6 Aluminum Alloys Joint and Post Heat Treatment"

_metals, doi:10.3390/met10010130_

Round 1

Reviewer 1 Report

Dear Authors, you presented in your work very interesting results related to the advanced welding technique, important for welding process optimization. Unfortunately, formulated conclusions were not confirmed by experimental results in a satisfactory degree. Therefore, they can only be regarded as qualitative. Some text corrections and supplementary analysis seem to be necessary.

- Text corrections :

Grammar mistakes and sentence style should be corrected Figures captions should be verified:

Fig.1 and Fig. 2 - the same captions

Fig. 5 - more detailed description is needed

Figs. 1, 7, 8 - expressions “optical microstructure” and “optical microscopy” should not be used for description of the microstructure state visible on micrographs

Fig. 8 -contains photos a-c, while in the text Figs 8d-h are mentioned

Some unclear parts of the text must be explained”

216 and next:

the results of microscopic examination in Figs.10 and 11 are not sufficient to support presented considerations regarding the degree of coherence and morphology of β″ phase precipitates

230 and next-

- it would be interesting to know the procedure of the measurement  used to obtain such results: „average width and length of them were about 22.73nm and 227.27nm, respectively”

 -what kind of precipitates was analysed and what did mean „size” in this case „precipitates the size of which was vary between 45.46 and 136.36nm”

- dispersion of the results should be given, especially since such occuracy was reached.

V238 and next

 what are the reasons for the statement : „Because of the feeding of ER5356 filler wire having high content of Mg element which provides avery favorable condition for the formation of β″ precipitate, the number of precipitates in the FZ was  larger than that in HAZ.”

To form β″ precipitate both elemnts Mg and Si are necessary, while the filler contains a lot of Mg but simultaneously very small amount of Si

Author Response

1) Fig.1 and Fig. 2 - the same captions

Answer: Fig.2 has been deleted.

2)Fig. 5 - more detailed description is needed

Answer: More detailed description has been added in the manuscript for Fig. 5.

The best parameters of the laser induced arc welding 3mm-thick Al alloys are shown in the Table 4. The welding appearance and macrostructure of the joint are shown in the Fig. 4. The welding joint was obtained in the free formation state. It could be found that the welding appearance was uniform and no hump defects. There was no welding porosity and the cracks in laser induced arc hybrid welding Al alloys joint. The width of welding beam was about 5mm on the top and 3mm at the bottom of the joint. The heat effect zone of the joint was less than 2mm.

3)Figs. 1, 7, 8 - expressions “optical microstructure” and “optical microscopy” should not be used for description of the microstructure state visible on micrographs

Answer: The captains of Fig.1, 7 and 8 were revised.

4)Fig. 8 -contains photos a-c, while in the text Figs 8d-h are mentioned Some unclear parts of the text must be explained”

Answer: The Fig. 8 has been changed and more details have been added in the manuscript.

5)L216 and next: the results of microscopic examination in Figs.10 and 11 are not sufficient to support presented considerations regarding the degree of coherence and morphology of β″ phase precipitates

Answer: In this process, the all results has shown in the manuscript. If you can give me more suggestion, I will do more experiment to explain it.

6)L230 and next-it would be interesting to know the procedure of the measurement used to obtain such results: “average width and length of them were about 22.73nm and 227.27nm, respectively” what kind of precipitates was analyzed and what did mean “size” in this case “precipitates the size of which was vary between 45.46 and 136.36nm”- dispersion of the results should be given, especially since such occuracy was reached.

Answer: I should say sorry for the precipitates size. It was obtained by measurement in the figures by the graphics software. Thus, it should be not so accuracy. The size of the precipitates was measured more carefully and mended as: Plenty of β″ were uniformly distributed in the FZ and the average width and length of them were about 10nm and 190nm, respectively. A great number of precipitates were uniformly dispersed in α (Al) matrix in the HAZ, whose size was vary between 25nm and 130nm. In the hybrid welding process, the Al alloy was fully melted, and the precipitate size was mainly decided by the solidification rate. However, in different parts of the laser-arc hybrid welding fusion zone, the solidification rate was obviously different, thus the size of precipitate was changed in extent.

7)V238 and next what are the reasons for the statement : “Because of the feeding of ER5356 filler wire having high content of Mg element which provides a very favorable condition for the formation of β″ precipitate, the number of precipitates in the FZ was larger than that in HAZ.” To form β″ precipitate both elements Mg and Si are necessary, while the filler contains a lot of Mg but simultaneously very small amount of Si.

Answer: In the solution and aging (STA) joint, according to the different conditions (temperature), precipitation phases with different morphologies and strengthening effects would be sequentially precipitated. The precipitation process is generally: supersaturated α solid solution → G • PI region (spherical objects without independent lattice structure) → G • PII region (Needle-like ordered structure β ″) → β ′ (rod-like semicoherent structure) → β (flaky equilibrium phase Mg2Si)

In the ER5356 filler wire, there was more Mg element in the fusion zone. Most of the Mg elements exist in the form of the equilibrium phase Mg2Si. Therefore more Mg2Si would be formed in the fusion zone. During the STA process, the number of alloying elements that can form a transition phase (β ″ and β ′) would be increased, which would helpful for the property of the welding joint.

Reviewer 2 Report

There is several undocumented claims in the paper, like: L118, with a claim that the heat input reduces the pores. (i dont think this is wrong - just undocumented)

Figure 6: Hardness continues to rise beyound the meassured distance from centerline - increase the length where the measurements are performed.

L172: Not sure everybody knows...

Include more tables of acieved results and measurements.

L147 (and other places): Too many decimals on numbers, the reuslts are too precise 

HAZ(soften zone) is stated many places, what does this mean? HAZ or is it another zone? State it only once, and the use the abbrevation (HAZ).

L174-184: Shortly describe the phases beta, beta' and beta'' to the reader, and what their difference are, and what their influence are on the results.

Fig10: additional descriptions, and add figures inducation where the details are zoomed from (e.g. fig b with a square on fig a).

L305 "Joint efficiency" is poorly described.

More details on the setup of Laser-hybrid welding, and the settings applied. What is the power (in W, not Amps) from the TIG?

Fig 4: Made images larger - there is plenty of space within the figure, so it does not make it larger.

Author Response

Answer to reviewer 2

1)Figure 6: Hardness continues to rise beyond the measured distance from centerline - increase the length where the measurements are performed.

Answer: The more hardness tests of the welding joint has been done, as shown in the Fig. 5.

2)L172: Not sure everybody knows...Include more tables of achieved results and measurements.

   Answer: The first sentence of the paragraph has been deleted.

3)L147 (and other places): Too many decimals on numbers, the results are too precise

Answer: The numbers in the manuscript have been changed into integer. As shown blow: Plenty of β″ were uniformly distributed in the FZ and the average width and length of them were about 10nm and 190nm, respectively. A great number of precipitates were uniformly dispersed in α (Al) matrix in the HAZ, whose size was vary between 25nm and 130nm.

4)HAZ (soften zone) is stated many places, what does this mean? HAZ or is it another zone? State it only once, and the use the abbreviation (HAZ).

   Answer: There would be a soften zone in the Arc welding 6061 Al alloy, because of the enhanced phase precipitation. The distance of this area was a little far from the common heat effect zone. Thus in the manuscript, it was called soften zone, which was used to distinguish the general heat affected zone and softened zone.

   As it made a misunderstanding effect, we have deleted the soften zone in the manuscript.

5)L174-184: Shortly describe the phases beta, beta' and beta'' to the reader, and what their difference are, and what their influence are on the results.

Answer: The explanation of beta, beta' and beta'' was added in the manuscript. In the solution and aging (STA) joint, according to the different conditions (temperature), precipitation phases with different morphologies and strengthening effects would be sequentially precipitated. The precipitation process is generally: supersaturated α solid solution → G • PI region (spherical objects without independent lattice structure) → G • PII region (Needle-like ordered structure β ″) → β ′ (rod-like semicoherent structure) → β (flaky equilibrium phase Mg2Si)

6)Fig10: additional descriptions, and add figures indication where the details are zoomed from (e.g. fig b with a square on fig a).

Answer: The captain of Fig. 9 has been changed.

7)L305 "Joint efficiency" is poorly described. More details on the setup of Laser-hybrid welding, and the settings applied. What is the power (in W, not Amps) from the TIG?

Answer: The "joint efficiency" has been deleted. It was revised as "The tensile property of STA joint was 84.05% of the base metal,"

8)Fig 4: Made images larger - there is plenty of space within the figure, so it does not make it larger.

The Fig. 4 is enlarged in the manuscript.

Reviewer 3 Report

I appreciate the way the information is organized. Makes the communication easier to follow, thanks.

I have a few comments on some of the other aspects including:

What are the parameters and the range of values investigated for each in this experiment? I cannot find that information. While some images point to some of them, the ideal parameters seem to indicate data that is not included. A table and/or a short discussion of the range of each of the parameters investigated and rationale behind their choice would be appreciated.

Discussion in lines 112-121 has to be improved. I do see the decrease in weld pores, but the discussion about the causality is very vague and confusing.

There are other typos across the document. Please revise.

Were the optical images in Figure 4, compiled by stitching multiple images together? If yes, then this needs to be specified along with details of the same somewhere in the discussion. It also looks like there are some overlap mismatches. For example, in the 100A arc current image in figure 4a, when looking from left to right, there are two regions, one at the end of image 2 and another at the beginning of image 3 that are very similar/identical. If you think they are not, please rebut. If you think they are identical please fix.

The resolution of the images in figure 4 also seems low. Can the authors please improve the image quality?

There are passages in the discussion section, that can be improved for clarity. Especially the sections detailing dislocation density, distribution and type changes. Please revise.

"there is no appreciable variation in the grain size of the joints for the PWHT methods evaluated". Any claims about variation have to supported by quantitative data. I do not see any statistically quantified data on grain sizes. Please provide the data or rephrase and/or remove inferences about grain sizes across the document. Since you have data on the sizes of the precipitates, having a table detailing the sizes of grains along with the above, before and after heat treatment will support your argument better.

Author Response

Answer to reviewer 3

1) What are the parameters and the range of values investigated for each in this experiment? I cannot find that information. While some images point to some of them, the ideal parameters seem to indicate data that is not included. A table and/or a short discussion of the range of each of the parameters investigated and rationale behind their choice would be appreciated.

Answer: We performed orthogonal experiments on the parameters, in order to get the best welded joint. This part included a lot experiments, so it is not added in the essay. In laser-arc hybrid welding process, it could be found that arc current was the main factor, followed by welding speed, and the influence of laser power was small, when laser beam power was larger than 800W.

The orthogonal test adjustment was mainly welding forming. This essay is mainly about the performance improvement and strengthening mechanism after heat treatment. Thus it was not discussed in the manuscript.

2) Discussion in lines 112-121 has to be improved. I do see the decrease in weld pores, but the discussion about the causality is very vague and confusing.

Answer: More discussion are added in the manuscript.

3) There are other typos across the document. Please revise.

Answer: The manuscript has been revised throughout.

4) Were the optical images in Figure 4, compiled by stitching multiple images together? If yes, then this needs to be specified along with details of the same somewhere in the discussion. It also looks like there are some overlap mismatches. For example, in the 100A arc current image in figure 4a, when looking from left to right, there are two regions, one at the end of image 2 and another at the beginning of image 3 that are very similar/identical. If you think they are not, please rebut. If you think they are identical please fix. The resolution of the images in figure 4 also seems low. Can the authors please improve the image quality?

Answer: Yes. This figure is compiled by stitching multiple images together. Because the area of the observation is too large. As the brightness and the point of the view of each images are different, thus it look like some difference. We have mend the figures as we could. However, it still not look very well. If it is not accepted, I will observe the joint again.

5) There are passages in the discussion section, that can be improved for clarity. Especially the sections detailing dislocation density, distribution and type changes. Please revise.

Answer: I should say sorry for this part. I do not know how revise this section. I really hope that you can help me.

6) "there is no appreciable variation in the grain size of the joints for the PWHT methods evaluated". Any claims about variation have to supported by quantitative data. I do not see any statistically quantified data on grain sizes. Please provide the data or rephrase and/or remove inferences about grain sizes across the document. Since you have data on the sizes of the precipitates, having a table detailing the sizes of grains along with the above, before and after heat treatment will support your argument better.

Answer: The sentence is deleted in the manuscript.

Reviewer 4 Report

Interesting paper on hybrid welding of Al alloys with a filler material. The manuscript can be improved after revision of the manuscript as suggested below:

“welding pole” à welding pool?

The advantages of TIG and laser welding for joining advanced materials should be added to the manuscript. See for example: 10.1016/j.jmatprotec.2019.03.020, 10.1016/j.matdes.2018.11.053 and 10.1016/j.optlastec.2017.09.038 and revise accordingly.

One critical issue often found in welding of Al alloys is porosity (see 10.1016/j.optlastec.2018.09.036). This should also by discussed in the introduction as it is rather important for structural applications.

How was the filler wire selected?

In fig 2 it looks like some words were deleted inside the figure. Please address.

Can the authors present the typical tensile curves for the BM and welded joints (with and without HTT)? It is important to evaluate the mechanical behaviour after welding.

The conditions for the microstructure changes in the fusion zone have been previously detailed in key review works 10.1007/s11837-003-0134-7 and 10.1016/j.pmatsci.2019.100590. This can support the microstructural discussion in section 3.4

For the TEM images leave some blank space between each photo to distinguish them clearly.

Very nice TEM work done.

Author Response

1) “welding pole” à welding pool?

Answer: “welding pole” is welding porosity, it has been mended.

2) The advantages of TIG and laser welding for joining advanced materials should be added to the manuscript. See for example: 10.1016/j.jmatprotec.2019.03.020, 10.1016/j.matdes.2018.11.053 and 10.1016/j.optlastec.2017.09.038 and revise accordingly. One critical issue often found in welding of Al alloys is porosity (see 10.1016/j.optlastec.2018.09.036). This should also by discussed in the introduction as it is rather important for structural applications.

Answer: The references are added in the manuscript.

3) How was the filler wire selected?

Answer: In the ER5356 filler wire, there was more Mg element in the fusion zone. Most of the Mg elements exist in the form of the equilibrium phase Mg2Si. Therefore more Mg2Si would be formed in the fusion zone. During the STA process, the number of alloying elements that can form a transition phase (β ″ and β ′) would be increased, which would helpful for the property of the welding joint.

4) In fig 2 it looks like some words were deleted inside the figure. Please address.

Answer:  Fig.2 has been deleted.

5) Can the authors present the typical tensile curves for the BM and welded joints (with and without HTT)? It is important to evaluate the mechanical behaviour after welding.

Answer: We cannot find the tensile curves for the BM and welded joints for the tensile tests in this manuscript. We must promote these experiments after one or two weeks.

6) The conditions for the microstructure changes in the fusion zone have been previously detailed in key review works 10.1007/s11837-003-0134-7 and 10.1016/j.pmatsci.2019.100590. This can support the microstructural discussion in section 3.4

Answer: The references are added in the manuscript.

7) For the TEM images leave some blank space between each photo to distinguish them clearly.

Answer: Some space is added between each photo in Fig.9 and 10.

Round 2

Reviewer 1 Report

Dear Authors,  the results of the precipitates  measurement and  precipitations sequence  are based for your conclusions .

However, the most important questions about methods of measuring  of the precipitates  have not been satisfactorily explained and remain for me still unclear.
The results of the microscopic examintations  have not been completed to properly support the presented conclusions.  Thus, I propose to consider one more correction before manuscript publication.

Author Response

Dear reviewer.

Your suggestion is very well. 

And We add the spectral analysis results of precipitated phases, as shown in the Fig.9. However, there are lots of precipitated phases, and we can only show one of them. We know it is still not enough for the measuring of the precipitates. If you have any  new suggestion, you can tell us.

Thank you for your hard work.

Reviewer 3 Report

While I understand that you dont think the experimental design warrants discussion in this article, the ideal parameters you have chosen lie outside of the experimental data you have presented in this article. There has to be a rationale provided for why a factor was chosen, why its range was chosen and some supporting evidence that the ideal parameters identified lie within the investigated experimental design space. While your work should be the basis of this argument, literature can also be provided in support of this argument.

Image quality is still poor. It is difficult to differentiate between pores and other macro/micro structural features. Also, my concern with the images is that when assessing weld porosity, the images for the investigation have to be accurately aligned, if not they may reflect higher or lower porosity than might be present. The 100A arc current in figure 3a, the 2nd and 3 image are not aligned correctly. The zone at the end of the image 2 is the same as the zone in the beginning of image 3. This zone has higher pore count than other regions in that image. This may increase the pore count than what is actually present. Please recheck the image and other images for these mismatches before assessing for weld porosity.

In figure 3b, using 430W laser power, the image shows macrostructural features that look like cracks (at the end just above the scale bar). It also seems that similar features might be present in other images, but because of the poor image quality, they are hard to identify. While these features may or may not be cracks, I am unable to tell whether the macro and micro structure is good or bad with these images. 

Author Response

Dear reviewer

You have privoded several constructive suggesitons.

However, we could not mend the Fig. 3 very well, thus we decided to delete this figure. Still,the information in Fig.3 is not very clear

And you suggestted that the welding parameters were important for the welding process. Therefore, we added partial welding parameters and the welding joint morphology. We hope the revisment could make helpful effect on the manuscript.  

Reviewer 4 Report

The manuscript was greatly improved.

Author Response

Dear reviewer

The conclusions of the manuscript has been mended.

Thank you very much

Round 3

Reviewer 1 Report

-

Author Response

Dear editor:

Thanks a lot.

I will sent the revision as soon as possible.